# Role of Bcl-2 Family Proteins in Photodynamic Therapy Mediated Cell Survival and Regulation

**DOI:** 10.3390/molecules25225308

**Published:** 2020-11-13

**Authors:** Eric Chekwube Aniogo, Blassan Plackal Adimuriyil George, Heidi Abrahamse

**Affiliations:** Laser research Centre, Faculty of Health Sciences, University of Johannesburg, P. O. Box 17011, Doornfontein 2028, South Africa; ericaniogo@gmail.com (E.C.A.); blassang@uj.ac.za (B.P.A.G.)

**Keywords:** Bcl-2 family, pro-apoptotic, anti-apoptotic, photodynamic therapy, cancer, cell survival

## Abstract

Photodynamic therapy (PDT) is a treatment modality that involves three components: combination of a photosensitizer, light and molecular oxygen that leads to localized formation of reactive oxygen species (ROS). The ROS generated from this promising therapeutic modality can be lethal to the cell and leads to consequential destruction of tumor cells. However, sometimes the ROS trigger a stress response survival mechanism that helps the cells to cope with PDT-induced damage, resulting in resistance to the treatment. One preferred mechanism of cell death induced by PDT is apoptosis, and B-cell lymphoma 2 (Bcl-2) family proteins have been described as a major determinant of life or death decision of the death pathways. Apoptosis is a cellular self-destruction mechanism to remove old cells through the biological event of tissue homeostasis. The Bcl-2 family proteins act as a critical mediator of a life–death decision of cells in maintaining tissue homeostasis. There are several reports that show cancer cells developing resistance due to the increased interaction of the pro-survival Bcl-2 family proteins. However, the key mechanisms leading to apoptosis evasion and drug resistance have not been adequately understood. Therefore, it is critical to understand the mechanisms of PDT resistance, as well as the Bcl-2 family proteins, to give more insight into the treatment outcomes. In this review, we describe the role of Bcl-2 gene family proteins’ interaction in response to disease progression and PDT-induced resistance mechanisms.

## 1. Introduction

Apoptosis is a self-limiting mechanism of cell death in which unwanted cells are removed in a biological manner, to maintain tissue homeostasis. This mechanism and its regulation are essential for normal cellular homeostasis and to prevent diseases like cancer from developing. In late 1980s, scientists successfully used molecular techniques to identify some machineries involved in apoptosis signal transduction pathway [1]. The process of apoptosis involves the permeabilization of the mitochondria membrane by the Bcl-2 family proteins, to release cytochrome c, which binds to the apoptotic protease-activating factor 1 (APAF1), to activate the caspase proteins that lead to cell death [2,3].

It has been well-established that photodynamic therapy (PDT) induces apoptotic cell death by changing the cell shape and structure. However, the cytoskeletons have been involved in the development of PDT resistance through signal transduction processes [4]. PDT-treated cells have shown a structural cytoskeleton reorientation that contributes in processes linked to proliferation and cell growth that mediate resistance. The mechanism of PDT mediated cell death strongly influences the impact of photo-damage to the cytoskeleton. These observations were seen in cell-death process after PDT, in which essential cytoplasmic actin binding protein to be underexpressed similar to what was already found for radio- and chemotherapy [5,6].

## 2. The Bcl-2 Family Proteins

The Bcl-2 is an oncogene that was identified in human B cell follicular lymphoma. The involvement of Bcl-2 in the apoptotic process was identified by Vaux and his co-workers in 1988 [7]. Bcl-2 family of protein is composed of three major groups of structurally related proteins that regulate apoptosis. These proteins are as follows: (1) the BH3-only proteins that sense and activate; (2) the pro-apoptotic and executioner proteins of Bax that permeabilize the mitochondrial outer membrane (MOM) and release cytochrome c; and (3) the antiapoptotic Bcl-2 members that will inhibit the executioner proteins and overhaul the process (Figure 1). The Bcl-XL and Bcl-w, a subfamily members of Bcl-2, share the same sequence homology, i.e., from BH1 to BH4 [8,9]. Meanwhile, the pro-apoptotic Bax and Bak member proteins share a sequence homology at BH1, BH2 and BH3, except BH4. Another subfamily of the pro-apoptotic proteins that share a sequence homology at the BH3 site are the Bik and Bid protein [10].

Proteins included in the Bcl-2 protein family are based on sequence homology of the founding member Bcl-2, in the four homologies of the BH region. Most members have a multiple binding membrane region on their carboxyl termini that facilitates binding of the proteins, and its location is on the MOM or to the endoplasmic reticulum (ER) membrane [11,12,13]. The BH4, together with BH1, BH2, and BH3 regions, is required for anti-apoptotic activity. Proteins like Bag-1, Raf-1 and calcineurin also modulate Bcl-2 activity by binding to the N-terminal BH4 domain region, to mediate anti-apoptotic function. The BH3 region has an affinity for Bax and Bak, which facilitate apoptotic cell death induction [14,15]. Furthermore, recent findings have highlighted the dual activities of the BH3 region in death agonist and antagonist manner. This means that the BH3 region of Bax can also bind to the Bcl-2 in a competitive manner, depending on their concentration, suggesting sequence similarities in the BH3 domain of the anti-apoptotic and apoptotic member proteins [10].

The multifaceted functions of Bcl-2 family members in cell death promotion and survival interaction mean that Bax can increase mitochondrial membrane permeability to release cytochrome c protein in response to an apoptotic stimuli [16]. Moreover, the Bcl-X_L_ competes with Bax at the BH3 homology site, to favor the pro-survival Bcl-X_L_ binding, thus causing the opposite of the first process [16]. In aged cells, this process is revered, and an increase in the pro-apoptotic proteins is favored to maintain tissue homeostasis [17].

## 3. Functions and Regulation of Bcl-2 Family Protein

The Bcl-2 family of proteins functions through mitochondrial binding interactions and permeability of its outer membrane, to regulate and control cell death activity. The regulation of MOM that favors the pro-apoptotic proteins family often results in membrane permeabilization and pore formation through which cytochrome c proteins are released for caspase cascades’ activation, to dismantle and destroy the cells [18]. The regulation of Bcl-2 could be transcriptional, which controls its expression, or post-translational modification, through dephosphorylating and ubiquitination that maintain its function [19].

Many researchers have described a model that highlights the role of membrane as the locus of action of most Bcl-2 family proteins [13,20]. The BH3 domain-binding groove exists as a multi-domain Bcl-2 family member site that is critical for activation and inhibition of apoptotic function. Once apoptosis is triggered, the activator BH3-only proteins bind to the BH3 domain-binding groove in Bax/Bak and activate them through a series of conformational changes that will cause mitochondrial membrane pore opening [21]. This binding interaction on the BH3 domain is competitive and occurs at the outer membrane of the mitochondria. Thus, the outer lipid bilayer of the mitochondria serves as an important site that governs the fate of activation or inhibition of the Bcl-2 family proteins in the mitochondrial outer membrane permeabilization (MOMP) [18].

Similarly, Korsmeyer and colleagues have proposed a model called the “rheostat model”, stating that apoptosis is determined through the amount of pro-apoptotic and survival proteins activated in a cell [22]. This model was postulated when they discovered that the abundant Bcl-2 proteins, though less than Bax proteins, were not able to inhibit cell death. Ultimately, the fate of the cells was determined through the interaction of relative abundance, affinity to membrane, localization and conformation of the Bcl-2 family members in apoptosis [18]. This can be linked to the work of Wang and co-workers, who have linked the aberrant expression of Bcl-2 protein with death resistance of cancer as a result of increased production of hydrogen peroxide. Oxidative stress mediated by peroxides can downregulate the Bcl-2 proteins through the dephosphorylation process [23].

## 4. Death and Survival Functions of Pro-Apoptotic Bcl-2 Proteins

The pro-apoptotic Bcl-2 effector proteins are Bax and Bak, which contain the three BH domain and are the major inducers of MOMP [24]. Under pathological conditions, expression of Bax and Bak protein did not initiate cell killing, contrary to its pro-death developmental function. This finding was extensively analyzed in Bak-deficient mice, where Bak was found to either inhibit or enhance neuronal death, thus suggesting its dual function in death and survival processes [25]. Cells devoid of Bax or Bak encoding genes were still found to be susceptible to apoptosis, although a significant functional redundancy was observed. This also demonstrates that both Bax and Bak are required together for Bax- and Bak-mediated membrane permeabilization and other apoptotic processes [13,26].

## 5. Intramitochondrial Functions of Bcl-2 Family Proteins

The fundamental function of Bcl-2 family proteins and its interaction with the shape changes, morphology and organelle localization in the mitochondria has been reported. Rolland and colleagues found that there is a close link between Bcl-2 family proteins and the dynamics of the GTPases Drp1 and Mfn1/2 that mediate the mitochondrial outer membrane fission and fusion process [27]. This process was first discovered when Drp1 was found to promote Bax-induced mitochondrial fission and cell death but contrarily promote fusion in a healthy cell mitochondria [28]. Despite the similarities and interactions of Bcl-2 family proteins, the connection between each specific pro- and anti-death Bcl-2 protein outcome becomes different because of the stimuli that initiate each process. Bax fusion triggered the pore transition opening on the mitochondria that leads to necrosis [29]. Recent studies have indicated that Bcl-2 proteins can be affected by the structure and function of the mitochondria, although previous knowledge showed that they exert their apoptotic function exclusively at the outer mitochondrial membrane. These findings followed the intramitochondrial localization of Bcl-2 and Bcl-X_L_ within the inner membrane [30]. Another antiapoptotic protein member, MCL-1, has been shown to possess a bona-fide mitochondrial pre-sequence that mediates changes in the mitochondrial matrix [26].

## 6. Photodynamic Therapy and Drug Resistance

Photodynamic therapy is a treatment that combines the interaction of light and a photosensitizer in the presence of molecular oxygen, to generate ROS that cause oxidative damage [31]. The photosensitizer preferentially localizes in essential cellular components, subsequently dictating the damage and outcome of the PDT treatment. Mostly, PDT cytotoxicity directly effects cells and triggers different cell death pathways, like apoptotic, necrotic or autophagic responses, depending on the cell types [32]. Other damage could lead to tumor vasculature and inflammatory reaction that develops systemic immunity [33].

The photosensitizer that initiates the photochemical reaction always exists in the ground state, and upon specific wavelength light exposure, it absorbs light photon and becomes excited in singlet unstable state (^1^PS*). The ^1^PS* is unstable in the sense that it has a short half-life, and through fluorescence emission, it returns to the ground state or undergoes intersystem crossing to a long-lived triplet excited state (^3^PS*). This new excited triplet state can also return to the ground state through phosphorescence, or be combined with molecular oxygen in a photochemical reaction, to induce phototoxicity. In the photochemical reaction (type I), the ^3^PS* reacts directly by electron or hydrogen biomolecules transfer, to form peroxides, superoxide ions and hydroxyl radicals. Meanwhile, in (type II), the ^3^PS* undergoes chemical reaction with molecular oxygen (^3^O_2_) to produce the singlet oxygen (^1^O_2_). The ^1^O_2_ generated is a very toxic activated oxygen molecule that is the key to the photo-cytotoxicity reaction followed by the PDT treatment, because of its interaction with proteins, nucleic acids and lipid biomolecules [34]. The photo-destruction of the cancerous tissue/cells by the highly reactive ROS molecules has a multifactorial impact that can lead to the tumor destruction via apoptosis and/or autophagy and/or necrosis (Figure 2). One of the protective mechanisms to the cellular response of PDT is the upregulation of antioxidant haem oxygenase-1 against oxidative stress. This antagonizes PDT treatment by scavenging ROS production and combination treatment with cellular antioxidant inhibitors; hydroperoxide degradation will especially boost ROS accumulation and enhance PDT cytotoxicity in the treatment of resistant cells [35,36].

Generally, light sources like lasers, lamps and Light-Emitting Diodes (LEDs) are used in PDT treatments, in so far that the characteristic spectrum matches with the maximum absorption wavelength of the chosen PS. However, the lasers are most preferable because they precisely deliver the amount of light needed for PS activation and wavelengths specificity, unlike the LEDs, which are very difficult to use to deliver light in most anatomical areas [37].

## 7. Challenges of Drug Resistance and Photosensitizer Uptake

Multidrug resistance (MDR) is a phenomenon attributed to the overexpression of ATP-binding cassette (ABC) proteins that inhibit transport across membranes. It occurs when cancerous cell become resistance to one or two anticancer drugs with related chemical structures and mechanism of action. The ATP-binding cassette (ABC) transporter proteins (also known as efflux pumps) involved in MDR include p-glycoprotein (P-gp), multidrug resistance protein (MRP) and breast cancer resistance protein (BCRP) [38]. Photosensitizers (PSs) are light-absorbing molecules that initiate the photochemical reactions but are not consumed within the reaction. They usually have the unique physicochemical properties, such as high solubility, chemical purity and stability, long activation wavelengths, enhanced selectivity, penetration and rapid excretion from the cancer tissue [39]. The PSs’ ability to selectively accumulate in the tumor area is largely influenced by lymphatic drainage, low pH and increased expression of low-density lipoprotein receptor in proliferating cancer cells [40]. Most PS are classified on the basis of their chemical structure and purity, as well as their specific targeting ability not to generate toxic degradation products upon irradiation [41]. The most PDT-resistant variants studied have revealed and suggested that physical properties of PS contributed significantly in the MDR phenotype development [42]. Others have also evaluated the phototoxicity of a number of PS dyes in MDR cells, which showed good results [43]. The PDT-resistance mechanism is linked to PS altered uptake, efflux and transport within cancer cells [44]. Another report on human urothelial cells has also shown that the BCRP transporter protein was a major factor that prevented 5-aminolevulini acid (5-ALA)-induced protoporphyrin (PpIX) accumulation and facilitates its efflux. This report has emphasized the importance of ABC transporter proteins as a molecular determinant in PS selectivity and PDT outcome [38].

Furthermore, the studies by Jonker and colleagues have demonstrated that mouse with overexpression of ABCG2 transporter has the ability to pump out a photosensitizer PpIX and protect against its phototoxicity [45]. Their findings were also strengthened by the correlation report between ABCG2 expression and PDT resistance [46]. The reduced accumulation of pyro-pheophorbide, chlorin e6 and 5-ALA-induced PpIX in alveolar carcinoma cells limits their photosensitization ability, suggesting an increased efflux and resistance to treatment [44]. In other words, ABCG2-mediated resistance could be PS and cancer-cell-line specific [44]. Photodynamic diagnosis, a technique used for the cancer diagnosis, has also shown the reduced accumulation of metabolized photosensitizer PpIX from 5-ALA, due to decreased expression of protoporphyrinogen IX oxidase (PPOX) and coproporphyrinogen oxidase (CPOX) enzymes involved in PpIX synthesis [47,48]. It was observed in gastric cancer cells that accumulation of PpIX was due and associated with enzymes involved in the biosynthetic pathway, thus contributing to the resistance development of 5-ALA-induced PpIX photosensitizer mediated treatment [48]. It is worth knowing that the structure of the PS plays a vital role in PDT-mediated resistance. A study by Usuda and colleagues in human epidermoid carcinoma A431 cells revealed that cells treated with PII-PDT were found to be resistant, unlike the one with mono-l-aspartyl chlorin e6 (NPe6-PDT). Both PSs have different structure that decrease and increase the production of ROS [49]. Other factors, like hypoxia, were found to assist in the pro-survival mechanism of PDT-mediated resistance [50]. Therefore, hypoxia might also be a factor in the PDT inhibition process that causes decreased PS accumulation and ineffective treatment.

## 8. Expression of Bcl-2 Family Proteins after PDT Treatment

The activation of anti-apoptotic Bcl-2 family proteins has been observed in PDT-treated resistant cells [51], which resulted due to the photo-damage of the proteins. Most PS used in PDT treatment targets the mitochondria, lysosomal and/or the endoplasmic reticulum, which are central to the induction of different cell death mechanisms after PDT [52]. The apoptosis-regulating genes of the Bcl-2 family proteins in PDT-resistant cells have been studied by Shen and colleagues [53] and were found to be upregulated together with heat-shock protein 27 (HSP27). In addition, there was a reduction in the mutant p53 protein in the resistant subline of the HT29 cells that indicates the Bcl-2 protein involvement in resistant PDT variant cells.

This involvement was supported by the protein expression level and its ability to suppress apoptosis in PDT-treated HL60 cells. The overexpressed anti-apoptotic protein blocks the release of the cytochrome c and activation of caspase proteins from the mitochondria [54]. The determinant of PDT response to either apoptotic stimuli or other forms of cell death has also been controversial. To find the usefulness of Bcl-2 in fate-directed PDT response, Koukourakis and colleagues investigated the much-debated role of Bcl-2 in PDT cell resistance with biopsy from esophageal cancer. They treated the biopsy with PDT, and their results showed a favorable degradation of Bcl-2 proteins that lead to apoptosis [55]. Xue et al. found that Pc4-mediated PDT can result in photo-damage of Bcl-2 in different human cell lines [56]. This Bcl-2 damage and sensitivity of the cancer cells to apoptosis is dependent on the Bcl-2/Bax ratio. Furthermore, higher doses of PDT were needed to change the conformational rearrangement of the pro-apoptotic Bax protein caused by overexpression of Bcl-2 protein [57].

As a tumor suppressor protein, p53 is the most common gene expressed in many tumors [44]. Its overexpression leads to many gene transcription that influence tumor responsiveness to therapy [58,59]. Once there is genetic instability or damage, p53-deficient cells fail to be regulated and result in continued tumor proliferation and progression [44]. Normally, the cells lacking p53 are less responsive to chemotherapy and are prone to the development of cancer resistant strain. It has also been shown that cells transfected with p53 proteins were more sensitive to apoptosis after PDT treatment than those in which the p53 genes were deleted or inactive [60]. In a biopsy screening of patients treated with PDT, there was no association of the p53 protein in the response [55,61]. In contrast, p53 protein aided in porphyrin-PDT-mediated cell death, as observed by Zawacka-Pankau and co-workers. They concluded that p53-mediated PDT cytotoxicity and direct interaction with PS drugs through accumulation and induction of p53-dependent cell death upon irradiation [62].

Cancer cell death mechanism after PDT can be classified as apoptotic and non-apoptotic, such as autophagic and necrotic cell death, depending on the type of stimuli and cellular damage [63,64]. Apoptosis is the cell death that involves activation of hydrolytic enzymes, leading to nuclear chromatin condensation and DNA fragmentation with morphological characterization, such as cell shrinkage and plasma membrane blebbing [65]. Necrotic cell death is characterized by cell swelling (oncosis), extensive plasma membrane damage, swelling of the cytoplasmic organelles and moderate chromatin condensation [66]. Autophagy is notable for its dual role in cell survival and cell death. PDT treatment activates proteins that trigger mitochondria-mediated apoptotic signaling pathways.

## 9. Mitochondria-Mediated Apoptosis

Apoptotic responses to PDT are carried out primarily in the mitochondria, through the intrinsic pathway [67]. It occurs mainly when photosensitizers, like photofrin and phthalocyanine derivatives, that localize in the mitochondria are used. The intrinsic apoptotic pathway is induced by diverse developmental and environmental factors, such as cellular stress, DNA damage, nutrient deprivation and cytotoxic insult [68]. The first response of the pathway is the disruption of mitochondrial membrane potential and release of apoptogenic proteins into the cytosol. These proteins include cytochrome c, endonuclease G (EndoG) and apoptosis-inducing factor (AIF). The cytochrome c undergoes conformational changes to form a complex called apoptosome and activates the hydrolytic caspase enzymes (Figure 3). Meanwhile the AIF and EndoG translocate from the mitochondria to the nucleus, where they mediate chromatin condensation and high-molecular-weight DNA fragmentation [68,69,70]. The caspases are a family of cysteine protease enzymes that are involved in the central apoptotic processes. Most apoptotic signaling process converges in the caspase activation, which is promoted through the cleavage of a wide range of cellular substrates, ultimately leading to packaging of the dying cells and their engulfment by phagocytes [68,71]. However, caspase 3 present in cells is activated by the cleavage of an upstream caspase or protease, such as granzyme B. This executioner caspase has been extensively studied and found to play a central role in both intrinsic and extrinsic cell death pathways, such that the cells deficient of it are protected from the final phase of apoptotic action [72]. Studies have shown a link between caspase 3 expression with lower survival rates in many cancer types, such as gastric cancer, ovarian cancer, cervical cancer and colorectal cancer [73]. PDT and many other anticancer therapies rely on promotion of apoptosis via caspase 3, and when this protease enzyme is compromised by mutation or deregulation, the outcome of treatment will be reduced. Downregulation or mutation within the caspase family can lead to incomplete activation of caspases and may contribute to chemoresistance and resurgence of tumor growth [72].

The membrane of the outer mitochondria is regulated by the Bcl-2 family of regulatory proteins; thus, they are the central regulators of the cell death pathway implicated in PDT-induced apoptosis [67]. Studies have found that antiapoptotic proteins are especially sensitive to endoplasmic reticulum (ER) and thus the mitochondria-targeted photodamage is the main functional mediator of PDT-induced apoptotic pathways [74]. Several human cancer cell lines loaded with phthalocyanine PS have been reported to cause photodamage with over expression of Bcl-2 family proteins, which enhances the apoptotic response due mitochondria disruption [75].

## 10. Death-Receptor-Mediated Apoptosis

Death-receptor-mediated apoptotic pathway occurs when photosensitizers targeting the cell membrane are used. The PS activation triggers specific cell death ligands from the tumor necrosis factor (TNF) superfamily (e.g., Fas ligand (FasL) or TNF-α), which interacts with their cell-surface receptors (Fas or TNF receptor-1 (TNFR1)) in the death domain of the cytoplasmic tail. One major receptor of the family is the Fas receptor, which is actively involved in PDT-induced apoptosis [75]. The Fas-associated death receptor forms a complex with FAS-associated death domain (FADD) protein and procaspase 8, together called death-inducing signaling complex (DISC). Upon proteolytic cleavage, caspase 8 activates other downstream effector caspases (caspase-3,6,7) [76,77]. Accordingly, this caspase activation is independent of cytochrome c release and mitochondrial membrane permeabilization, which drives the intrinsic apoptotic pathway [74]. However, caspase 8 activated by death receptor signal can form a complex with Bid which rapidly translocate to the mitochondria and binds with the Bax or Bak of the Bcl-2 family of proteins and initiates apoptosis through caspase cascade. This way, the extrinsic pathway can shift to intrinsic pathway through Bid activation, by caspase 8, to achieve apoptosis. Bid has been described as being dispensable for apoptosis, despite being the molecular connector between intrinsic and extrinsic apoptotic pathways [78,79,80]. Another similar report has stated that Bid serves as a guard for the mitochondrial pathway for apoptosis, but how it activates the Bax and Bak proteins is still unclear [81]. PDT-induced death-receptor-mediated apoptosis has been observed in many cell lines with Hypericin PS, in different experimental settings [32,67,75]. Similarly, both in vitro and in vivo PDT experimentation has been reported to increase the expression of Fas death receptor and its ligand (FasL) [32,76,82]. Such overexpression was also observed in tumor-bearing mice, and the formation of Fas-FADD complexes and activation of caspase-8 were also observed following the PDT, suggesting the apoptosis via death-receptor-mediated signaling [83,84]. PDT-induced apoptosis can also be influenced by other cellular signaling pathways, like calcium homeostasis, ceramide formation and MAPKs [75,85,86].

## 11. Bcl-2 Family Proteins in Autophagic Response to Photodynamic Therapy

PDT induces cell death either by apoptosis or a non-apoptotic manner, depending on the cell type, photosensitizer localization and light fluency [76]. In the apoptotic pathway previously explained, phototoxicity in the mitochondrial will trigger a caspase cascade reaction that kills the cell. However, when the PS localizes in the lysosome, PDT toxicity will lead to photo-degradation of lysosomal enzymes and, thus, can lead to gradual dying of the cell. The later response can lead to either death or survival of the specific stimulation. When ROS production is not adequate to cause damage to the lysosome, especially in cells found to be lacking Bax and procaspase-3 enzyme protein, autophagy is said to be activated in response to cell survival and proliferation. Accordingly, Xue and colleagues [87] observed that cells die slowly once apoptosis is activated in Bcl-2 overexpressed cells as to what is initially known of protein protection against apoptosis. The Bcl-2 anti-apoptotic protein did not protect the cells from phototoxicity [88]. Later, it was reversed that the upregulation of the Bcl-2 protein in the same PDT-treated cells provided substantial protection against reproductive-cell death [57]. In other words, showing that Bcl-2 does not always protect against PDT phototoxicity. Previous reports have demonstrated that PDT-induced autophagy could be a common phenomenon that occurs earlier than apoptosis and is independent of pro-apoptotic proteins [89,90]. Cells preferentially die by apoptosis, or through cytoplasmic content sequestration regulated by type I and III phosphoinositol-3-kinase proteins that can antagonize apoptosis [91]. In cases where both apoptosis and autophagy were induced in response to PDT, pro-apoptotic capacity will dominate, and Bcl-2 proteins will protect against the process, to enhance autophagy flux and response [87]. Murine-resistant embryonic cells with a double knockout Bax/Bak protein after PDT treatment underwent a non-apoptotic cell death (type II autophagy) enhanced by the overexpression of Bcl-2 or Bcl-xL proteins [92]. This process was in opposition to the report of Pattingre and Levine [93]; they demonstrated the Bcl-2 protection and binding to the Beclin-1 autophagic protein. Their findings stated that Bcl-2 blocks autophagic cell death by binding to Beclin 1, which inhibits autophagy [93]. It is possible that PDT-induced Bcl-2 photodamage disrupts the binding of Bcl-2 to Beclin 1, resulting in elimination of Bcl-2 effect on PDT-induced autophagy. This response contributes to cancer-treatment-related resistance mechanism, and studies have been conducted to identify a potential strategy to enhance sensitivity of cancer cells to PDT. Wei and colleagues identified that inhibition and silencing of autophagic *ATG5* gene by pharmacological inhibitors substantially triggered apoptosis in resistant tumor cells [94]. Pharmacological inhibitors like 3-Methyladenine, bafilomycin A1, obatoclax, clarithromycin, chloroquine and hydrochloroquine, etc., tend to target the inhibitory effect of cytoprotective autophagy to overcome therapy resistance [95]. Another study by Domagala et al. also confirms that *ATG5* gene knockout in HeLa and MCF-7 cells moderately increases the efficacy of photofrin-based PDT, thus suggesting involvement of the autophagy gene in PDT therapy [96]. This shows that autophagic genes need to be blocked or somehow be inhibited, in order to increase the efficiency of PDT therapeutic responses. A possibility is that ROS production from PDT could trigger nutrient deprivation and oxygen stress, a mechanism that contributes to autophagy and enables the cell to escape from damage. Hence, targeting autophagy with inhibitors in combination with PDT will enhance sensitivity and promote tumor cell death. This strategy has been supported by a recent report, as a smart emerging procedure that can annul the interference of autophagic response in cancer therapies [95].

## 12. Conclusions

In conclusion, the mechanisms through which Bcl-2 family proteins promote or antagonize cell death are still elusive and constantly attract research investigations. Reports have shown that most PSs localize in the mitochondria, which is vital for apoptotic induction. During PDT, the activated ROS causes damage to the mitochondria, which triggers a preferred cell-death mechanism. Nonetheless, the resistance mechanism of PDT is fast evolving and has been linked to PS-altered uptake and efflux within the cell. The apoptosis-regulating proteins of the Bcl-2 family in PDT-resistant cells, if profiled and analyzed, may hint at the molecular interactions that sustain cell survival after PDT, which might possibly offer an insight towards winning the risk of MDR in cancer treatment.

## 13. Future Perspectives

As discussed above, the cancer cells resist treatment through anti-apoptotic adaptation mediated by Bcl-2 family proteins. This have increased the interest of most researchers to find possible ways of addressing the problem and bringing a lasting solution to cancer-therapy resistance. More recently, scientists have developed small molecules called BH3 mimetics, to inhibit and mimic the action of certain BH3-only proteins. These drugs show relatively low affinity to Bax/Bak but target the surface pocket of Bcl-2, to abrogate its function and encourage cancer regression. Some examples of the Bcl-2 inhibitors developed (obatoclax (GX-15-070), ABT-737, Navitoclax and venetoclax) have been found to be useful in predisposing cancer cells to apoptosis [97]. Taking advantage of these biological BH3 mimetics drugs, in combination with photodynamic therapy, undoubtedly increases the efficacy of the therapy by blocking anti-apoptotic proteins and prevent resistance. However, in this early stage of BH3 mimetics development, there might be a challenge on when and the best inhibitor molecule to use, since there are a dozen members of Bcl-2 family proteins. These proteins can undergo post-translational modification and protein-to-protein interactions that will make difficult the correct use of BH3 mimetics to use. The high off-target side effects of the BH3 mimetics have been of interest since they limit the efficacy and results to dose-limiting toxicities. This challenge can best be managed by combining BH3 mimetics molecule with tumor-specific agents, to selectively attack and induce apoptosis in cancer cells. These strategies, if carefully implemented, might provide a successful route for the treatment of cancer resistance.

## Figures and Tables

**Figure 1 molecules-25-05308-f001:**
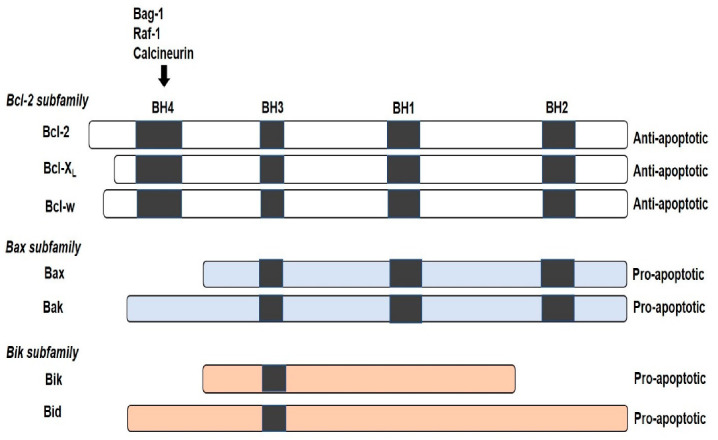
The homology of Bcl-2 family proteins. There are three subfamilies of Bcl-2 proteins related by structure and function, with four different sequence homology regions, designated as BH1, BH2, BH3 and BH4. The BH3 region is common in all the subfamilies and participates in both anti- and pro-apoptotic activities. Non-Bcl-2 family proteins like Bag-1, Raf-1 and calcineurin also bind to the BH4 domain region.

**Figure 2 molecules-25-05308-f002:**
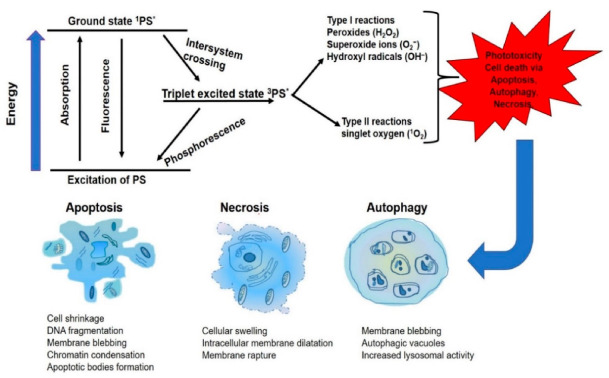
Mechanism of photodynamic therapy (PDT). The toxic effect of reactive oxygen species molecules (hydroxyl radicals (OH^−^)), peroxides (H_2_O_2_) and superoxides (O_2_^−^) produced via type I and type II (excited-state singlet oxygen (^1^O_2_)) photochemical processes that lead to phototoxicity and subsequent cell death via apoptosis, necrosis or autophagy.

**Figure 3 molecules-25-05308-f003:**
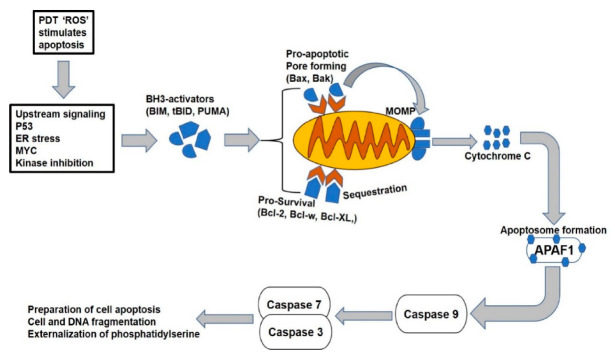
The mitochondrial apoptotic pathway. The pathway is initiated through cellular stress or damaged signals that unleash the BH3-only activators through either bound to pro-survival or pro-apoptotic Bax/Bak proteins. The latter causes the mitochondrial membrane pore opening and leads to the mitochondrial outer membrane permeabilization (MOMP), resulting in the release of apoptogenic molecules and subsequent activation of caspases to initiate the cell death.

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
