# Peer review of "Role of Bcl-2 Family Proteins in Photodynamic Therapy Mediated Cell Survival and Regulation"

_molecules, 2020, doi:10.3390/molecules25225308_

Round 1
Reviewer 1 Report
Submitted document is a revised version of a manuscript of a review on a role of proteins of the Bcl-2 family in regulation of cell survival in photodynamic therapy of cancer. In previous version I found many flows throughout the whole text and I specifically listed some of the most striking examples. While authors attempted to address these, I believe that these changes generally are not satisfactory.
(e.g. line 33-36 - There is no interaction between pro-apoptotic Bcl-2 proteins and cascades. Bcl-2 family proteins permeabilize the mitochondrial membranes to release cytochrome c from mitochondria. Cytochrome c then in cytosol binds to proteins (Apaf1) that activate caspases. In the manuscript, it looks like interaction of pro-apoptotic Bcl-2 proteins with caspases results in release of cytochrome c, which is indeed wrong.)
(lines 48-55 - This is not factually wrong, but it is confusing. First mentioned group of proteins is called BH3-olny proteins (not BH3 proteins). From the text it also appears like all members are mentioned, while these are only examples.)
(lines 57 -60 and 65-68 It would be better to omit the mention of Bag1, Raf1 and calcineurin in the figure. Mentioning them does not add anything to the text.)
In paragraph 72-78 - There is nothing controversial on the fact that Bax increases mitochondrial membrane permeability. It is well accepted that it does form pores in membrane. Rest of the paragraph appears so confusing that I do not understand it.
It appears that the majority of text remains untouched. Even though I did not specifically list the mistakes in the rest of the text, in the previous review I indicated that whole text must be edited for English an factual flaws. These are many little flaws that I am not capable to list all:
examples are: (paragraph 87-96)
- line 88: this is the first time mentioning BH3-bindig groove, must be indicates that this is present on multi-domains Bcl-2 family members
- stimulus is NOT initiated
- proteins bind (not proteins binds)
- inhibition of antiapoptoic proteins by BH3-only proteins is NOT reversal of activation of Bax/Bak by BH3-only proteins.
- last sentence in the paragraph is incomprehensible. Must be rephrased.
(lines 110-112) Levi-Montalcini (ref 26) did not found anything about Bax. Bax was discovered a quarter century later.
I am sorry to have to write again that while the topic is interesting and actual, this version of manuscript still fails in being a good review. Its English and writing style must be improved throughout the whole text.
This is not to discourage authors from publishing the review. I believe that after carefully rewriting the text, this may be interesting and useful review. I have, however, to recommend not to accept this manuscript for publication in the present form.
Author Response
|
Reviewer 1 Comments |
Authors Response |
|
Submitted document is a revised version of a manuscript of a review on a role of proteins of the Bcl-2 family in regulation of cell survival in photodynamic therapy of cancer. In previous version I found many flows throughout the whole text and I specifically listed some of the most striking examples. While authors attempted to address these, I believe that these changes generally are not satisfactory. |
Thanks for the comments. Authors have addressed all the comments/suggestions raised by the reviewer to improve the quality of the article and readership. |
|
(e.g. line 33-36 - There is no interaction between pro-apoptotic Bcl-2 proteins and cascades. Bcl-2 family proteins permeabilize the mitochondrial membranes to release cytochrome c from mitochondria. Cytochrome c then in cytosol binds to proteins (Apaf1) that activate caspases. In the manuscript, it looks like interaction of pro-apoptotic Bcl-2 proteins with caspases results in release of cytochrome c, which is indeed wrong.) |
The sentence has been rephrased for better understanding. Page 1; line 33 – 36. |
|
(lines 48-55 - This is not factually wrong, but it is confusing. First mentioned group of proteins is called BH3-olny proteins (not BH3 proteins). From the text it also appears like all members are mentioned, while these are only examples.) |
The sentence has been restructured for better understanding. Page 2; line 48 – 55. |
|
(lines 57 -60 and 65-68 It would be better to omit the mention of Bag1, Raf1 and calcineurin in the figure. Mentioning them does not add anything to the text.) |
Thanks for the input, but the authors feel it is also good to highlight other non-apoptotic proteins that modulate BH4 antiapoptotic domain site. |
|
In paragraph 72-78 - There is nothing controversial on the fact that Bax increases mitochondrial membrane permeability. It is well accepted that it does form pores in membrane. Rest of the paragraph appears so confusing that I do not understand it. |
The paragraph has been restructured for better understanding. Page 3; line 72 – 77. |
|
It appears that the majority of text remains untouched. Even though I did not specifically list the mistakes in the rest of the text, in the previous review I indicated that whole text must be edited for English an factual flaws. These are many little flaws that I am not capable to list all: |
Typo and grammatical errors have been corrected in the manuscript. |
|
examples are: (paragraph 87-96) line 88: this is the first time mentioning BH3-bindig groove, must be indicates that this is present on multi-domains Bcl-2 family members |
It has been corrected and included in page 3; line 87 – 88. |
|
stimulus is NOT initiated |
Corrected page 3; line 89 |
|
proteins bind (not proteins binds) |
Corrected page 3; line 89 |
|
inhibition of antiapoptoic proteins by BH3-only proteins is NOT reversal of activation of Bax/Bak by BH3-only proteins. |
The sentence has been corrected for better understanding. Page 3; line 91 – 92 |
|
last sentence in the paragraph is incomprehensible. Must be rephrased. |
The sentence has been corrected as suggested. Page 3; line 93 – 96 |
|
(lines 110-112) Levi-Montalcini (ref 26) did not found anything about Bax. Bax was discovered a quarter century later. |
The sentence has been deleted. Thanks |
|
I am sorry to have to write again that while the topic is interesting and actual, this version of manuscript still fails in being a good review. Its English and writing style must be improved throughout the whole text. This is not to discourage authors from publishing the review. I believe that after carefully rewriting the text, this may be interesting and useful review. I have, however, to recommend not to accept this manuscript for publication in the present form. |
The authors believe that all the suggested corrections have carried out and the revised version of the manuscript will benefit to the readership. |
|
Reviewer 2 Comments |
Authors Response |
Reviewer 2 Report
The authors provide a review of the role of Bcl-2 family proteins in photodynamic therapy (PDT), especially in the resistance of PDT. This review article seems to be informative for the researchers in this specific field. Some revisions are suggested as followed.
Major comments
- In Figure 2, the activation of caspase-3/7 should be the main reason for the therapeutic effect of PDT. It is known that some cancers harbor somatic mutation of caspase-3 (Hum Genet. 2004 Jul;115(2):112-5.). The authors should discuss the effect of CASP3 somatic mutations on the outcome of PDT.
- In the perspective section, the authors suggest that the combination of BH3 mimetics with PDT could be a solution for PDT resistance. However, one of the problems of BH3 mimetics in cancer treatment is the side effect (Integr Cancer Sci Therap. 3: DOI: 10.15761/ICST.1000184.). The authors should discuss how to avoid the side effects of BH3 mimetics when combined with PDT.
- ROS is suggested as the trigger for PDT-induced cell death in Figure 2 but the authors did not provide any comment or suggestion about the possible application in the inhibition of cellular antioxidants in PDT therapy.
Minor comments:
1. Many abbreviations came out without providing the full name when first appearance, such as MOMP (Line 96), BCRP (Line 185), PpIX (Line 186), and so on.
Author Response
|
Reviewer 2 Comments |
Authors Response |
|
The authors provide a review of the role of Bcl-2 family proteins in photodynamic therapy (PDT), especially in the resistance of PDT. This review article seems to be informative for the researchers in this specific field. Some revisions are suggested as followed. |
Thanks for the comments to improve the quality of article |
|
In Figure 2, the activation of caspase 3/7 should be the main reason for the therapeutic effect of PDT. It is known that some cancers harbor somatic mutation of caspase-3 (Hum Genet. 2004 Jul;115(2):112-5.). The authors should discuss the effect of CASP3 somatic mutations on the outcome of PDT. |
Explanation of the Caspase 3 and outcome of PDT have been included on page 7, line 264 – 273. |
|
In the perspective section, the authors suggest that the combination of BH3 mimetics with PDT could be a solution for PDT resistance. However, one of the problems of BH3 mimetics in cancer treatment is the side effect (Integr Cancer Sci Therap. 3: DOI: 10.15761/ICST.1000184.). The authors should discuss how to avoid the side effects of BH3 mimetics when combined with PDT. |
The suggestion has been included on page 9, line 379 – 382. |
|
ROS is suggested as the trigger for PDT-induced cell death in Figure 2 but the authors did not provide any comment or suggestion about the possible application in the inhibition of cellular antioxidants in PDT therapy. |
Authors have included the suggestion. Page 5, 159 – 163. |
|
Many abbreviations came out without providing the full name when first appearance, such as MOMP (Line 96), BCRP (Line 185), PpIX (Line 186), and so on. |
The expanded forms for the abbreviations have been included. Page 3, line 95, page 5, line 174 and 188 respectively. |
Reviewer 3 Report
The manuscript "Role of Therapy Mediated Cell Survival and Regulation Bcl - 2 Family Proteins in Photodynamic Therapy Mediated Cell Survival and Regulation" by the authors Eric Chekwube Aniogo, Blassan Plackal Adimuriyil George, and Heidi Abrahamse is very relevant for the development of the photodynamic therapy method and should be published.
With revision by the authors, the manuscript is now even better.
Nevertheless, it is necessary to clarify a few points again with PpIX.
P5, L189: instead “Furthermore, the studies by Jonker and colleagues have demonstrated that mouse with overexpression of ABCG2 transporter has the ability to pump out a prodrug of 5-ALA photosensitizer PpIX” please write “Furthermore, the studies by Jonker and colleagues have demonstrated that mouse with overexpression of ABCG2 transporter has the ability to pump out a photosensitizer PpIX”.
P5, L192: instead “The reduced accumulation of pyro-pheophorbide, chlorin e6 and 5-ALA PpIX photosensitizers in alveolar carcinoma cells limits their photosensitization ability which suggests increased efflux and resistance to treatment [44]” please write “The reduced accumulation of pyro-pheophorbide, chlorin e6 and 5-ALA induced PpIX in alveolar carcinoma cells limits their photosensitization ability which suggests increased efflux and resistance to treatment [44]".
P5, L195: instead “Photodynamic diagnosis, a technique used for the cancer diagnosis have also shown the reduced accumulation of metabolized PpIX from 5-ALA photosensitizer" please write “Photodynamic diagnosis, a technique used for the cancer diagnosis have also shown the reduced accumulation of metabolized photosensitizer PpIX from 5-ALA".
Author Response
|
Reviewer 3 Comments |
Authors Response |
|
The manuscript "Role of Therapy Mediated Cell Survival and Regulation Bcl - 2 Family Proteins in Photodynamic Therapy Mediated Cell Survival and Regulation" by the authors Eric Chekwube Aniogo, Blassan Plackal Adimuriyil George, and Heidi Abrahamse is very relevant for the development of the photodynamic therapy method and should be published. With revision by the authors, the manuscript is now even better. |
Thanks for the comments to improve the quality of article. |
|
Nevertheless, it is necessary to clarify a few points again with PpIX. P5, L189: instead “Furthermore, the studies by Jonker and colleagues have demonstrated that mouse with overexpression of ABCG2 transporter has the ability to pump out a prodrug of 5-ALA photosensitizer PpIX” please write “Furthermore, the studies by Jonker and colleagues have demonstrated that mouse with overexpression of ABCG2 transporter has the ability to pump out a photosensitizer PpIX”. |
Corrected ad s suggested. Page 5; line 192. |
|
P5, L192: instead “The reduced accumulation of pyro-pheophorbide, chlorin e6 and 5-ALA PpIX photosensitizers in alveolar carcinoma cells limits their photosensitization ability which suggests increased efflux and resistance to treatment [44]” please write “The reduced accumulation of pyro-pheophorbide, chlorin e6 and 5-ALA induced PpIX in alveolar carcinoma cells limits their photosensitization ability which suggests increased efflux and resistance to treatment [44]". |
The change has been implemented. Page 5; line 195. |
|
P5, L195: instead “Photodynamic diagnosis, a technique used for the cancer diagnosis have also shown the reduced accumulation of metabolized PpIX from 5-ALA photosensitizer" please write “Photodynamic diagnosis, a technique used for the cancer diagnosis have also shown the reduced accumulation of metabolized photosensitizer PpIX from 5-ALA". |
The changes were made as suggested. Page 5; line 198 and page 6; line 199. |
Round 2
Reviewer 1 Report
Submitted document is another revised version of a manuscript of a review on a role of proteins of the Bcl-2 family in regulation of cell survival in photodynamic therapy of cancer. In previous version I found many flows throughout the whole text and I specifically listed some of the most striking examples. While authors attempted to address these, I believe that these changes generally are not satisfactory. Here I list examles of my previous concerns that has NOT been addresed:
line 33-36 - There is no interaction between pro-apoptotic Bcl-2 proteins and cascades. Bcl-2 family proteins permeabilize the mitochondrial membranes to release cytochrome c from mitochondria. Cytochrome c then in cytosol binds to proteins (Apaf1) that activate caspases. In the manuscript, it looks like interaction of pro-apoptotic Bcl-2 proteins with caspases results in release of cytochrome c, which is indeed wrong.)
- I do not see any improvement. I nthe text it still appears like the interaction of pro-apoptotic Bcl-2 proteins with caspases woulld result in release of cytochrome c. This is wrong.)
lines 48-55 - This is not factually wrong, but it is confusing. First mentioned group of proteins is called BH3-olny proteins (not BH3 proteins). From the text it also appears like all members are mentioned, while these are only examples.)
- No improvement, (even the BH3 roteins has not been changed to BH3-only proteins - ln.49)
inhibition of antiapoptoic proteins by BH3-only proteins is NOT reversal of activation of Bax/Bak by BH3-only proteins.
- Again on improvement.
In my opinion the manusript, thus, does not describe the action of proteins of Bcl2 family as we know it. Other than that I do not see any improvement in English and writing style (which would be easily fixed, by giving the manuscript to a native English sepeaker for correction).
I am sorry but I must therefore recommend not to accept the manuscript for publication.
Author Response
|
Reviewer 1 Comments |
Authors Response |
|
Submitted document is another revised version of a manuscript of a review on a role of proteins of the Bcl-2 family in regulation of cell survival in photodynamic therapy of cancer. In previous version I found many flows throughout the whole text and I specifically listed some of the most striking examples. While authors attempted to address these, I believe that these changes generally are not satisfactory. Here I list examles of my previous concerns that has NOT been addresed: |
Thanks for the comments. Authors have addressed all the comments/suggestions raised by the reviewer to improve the quality of the article and readership. |
|
line 33-36 - There is no interaction between pro-apoptotic Bcl-2 proteins and cascades. Bcl-2 family proteins permeabilize the mitochondrial membranes to release cytochrome c from mitochondria. Cytochrome c then in cytosol binds to proteins (Apaf1) that activate caspases. In the manuscript, it looks like interaction of pro-apoptotic Bcl-2 proteins with caspases results in release of cytochrome c, which is indeed wrong.) - I do not see any improvement. I nthe text it still appears like the interaction of pro-apoptotic Bcl-2 proteins with caspases woulld result in release of cytochrome c. This is wrong.) |
The sentence has been modified. Page 1; line 33 – 35. |
|
lines 48-55 - This is not factually wrong, but it is confusing. First mentioned group of proteins is called BH3-olny proteins (not BH3 proteins). From the text it also appears like all members are mentioned, while these are only examples.) |
In that paragraph authors are trying to explain the structural representation of homologies and functional properties of Bcl-2 family proteins in figure 1. However, BH3-only proteins that share sequence homolog at BH3 sites are mentioned for clarity in page 2; line 55 – 56. |
|
No improvement, (even the BH3 proteins has not been changed to BH3-only proteins - ln.49) |
The change has been included Page 2; line 49 |
|
inhibition of antiapoptoic proteins by BH3-only proteins is NOT reversal of activation of Bax/Bak by BH3-only proteins. - Again on improvement. |
The sentence has been corrected Page 3; line 90 – 92. |
|
In my opinion the manusript, thus, does not describe the action of proteins of Bcl2 family as we know it. Other than that I do not see any improvement in English and writing style (which would be easily fixed, by giving the manuscript to a native English sepeaker for correction). I am sorry but I must therefore recommend not to accept the manuscript for publication. |
The manuscript has been checked and corrected to improve its readership. |
Reviewer 2 Report
The authors have improved their manuscript according to my previous suggestions. The manuscript is acceptable.
Author Response
Authors sincerely thank the reviewer for the meticulous comments to improve the quality of the manuscript.